# Enhancing Minds in Motion® as a virtual program delivery model for people living with dementia and their care partners

**Bobby Neudorf**[1], **Christopher Dinh**[1], **Vanessa Barnes**[2], **Christina Stergiou-Dayment**[2], **Laura Middleton**[1,3]*

**1** Department of Kinesiology, University of Waterloo, Waterloo, Ontario, Canada, **2** Alzheimer Society of Ontario, Toronto, Ontario, Canada, **3** Research Institute for Aging, Toronto, Ontario, Canada

* laura.middleton@uwaterloo.ca

**Data Availability Statement:** All relevant data are within the paper and its Supporting information files.

## Abstract

The Alzheimer Society of Ontario's Minds in Motion (MiM) program improves physical function and well-being of people living with dementia (PLWD) and their care partners (CP) (Regan et al., 2019). With the COVID-19 pandemic, there was an urgent need to transition to a virtual MiM that was similarly safe and effective. The purpose of this mixed methods study is to describe the standardized, virtual MiM and evaluate its acceptability, and impact on quality of life, and physical and cognitive activity of participants. Survey of ad hoc virtual MiM practices and a literature review informed the design of the standardized MiM program: 8 weeks of weekly 90-minute sessions that included 45-minutes of physical activity and 45-minutes of cognitive stimulation in each session. Participants completed a standardized, virtual MiM at one of 6 participating Alzheimer Societies in Ontario, as well as assessments of quality of life, physical and cognitive activity, and program satisfaction pre- and post-program. In all, 111 PLWD and 90 CP participated in the evaluation (average age of 74.6±9.4 years, 61.2% had a college/university degree or greater, 80.6% were married, 48.6% of PLWD and 75.6% of CP were women). No adverse events occurred. MiM participants rated the program highly (average score of 4.5/5). PLWD reported improved quality of life post-MiM (p = <0.01). Altogether, participants reported increased physical activity levels (p = <0.01) and cognitive activity levels (p = <0.01). The virtual MiM program is acceptable, safe, and effective at improving quality of life, cognitive and physical activity levels for PLWD, and cognitive and physical activity levels among CP.

## Introduction

In 2019, there was estimated to be over 50 million people living with dementia globally [1], which is projected to triple to 152 million people in 2050 [1–3]. Though dementia is defined by cognitive impairment sufficient to impede functional abilities, changes in physical function [4], sensory processing [5], communication [6], and mood [7] are also common. The effect of these diverse changes may compound to detrimentally impact functional abilities and quality of life among people living with dementia and their care partners [8–11]. For care partners,

**Funding:** 2122-HQ-000230 CA - Public Health Agency of Canada, received by C.S.D https://www.canada.ca/en/public-health.html The funders had no role in the study design, data collection and analysis, decision to publish, or preparation of the manuscript.

**Competing interests:** The authors have declared that no competing interests exist.

providing care can be rewarding [12], but may also have a negative impact on the care partner's health, work, and social life [1, 13–15].

Since the few pharmaceutical treatments for dementia fall short of a cure, identification of alternative strategies to promote health, function, and well-being of people living with dementia is important [16, 17]. Exercise is recommended as a key component of managing dementia [16, 17]. Exercise (i.e., physical activity done for the purpose of improving fitness and function) improves cognitive function, mobility, endurance, strength, balance, and mood among people living with dementia [18–23]. Though the most studies have focused on aerobic exercise, combined aerobic and resistance exercise may have greater benefits than either alone [24]. Exercise interventions may help to maintain independence and so enable people living with dementia to continue living well, while limiting their dependency on others [25–27]. Care partners also benefit from exercise, as it reduces carer strain and can improve endurance and strength, support quality of life, mood, and confidence, and reduce fall risk [19, 28–33].

Cognitive stimulation is another non-pharmaceutical intervention that may support the function of people living with dementia. Cognitive stimulation therapy includes a variety of enjoyable activities that stimulate thinking, concentration, and memory, often in a small-group environment [17]. Some clinical management guidelines recommended group cognitive stimulation therapy as an intervention for individuals living with mild to moderate dementia [17, 34]. Preliminary evidence suggests that cognitive stimulation therapy may improve quality of life and functional abilities while reducing depressive symptoms [35–38]. Given the broad benefits of physical activity and cognitive stimulation, it is reasonable to expect that a combined program of physical activity and cognitive stimulation may improve the well-being of people living with dementia and care partners.

Minds in Motion® (MiM®) was introduced by the Alzheimer Society of British Columbia (Canada) in 2012, adapted and offered by the Alzheimer Society of Ontario in 2014 [39], and has since expanded across most of Canada. MiM® includes exercise and cognitive stimulation in a social, group environment. In an evaluation of nearly 900 people, MiM® participants experienced improvements to physical function, physical activity, and perceived well-being [39].

Public health restrictions introduced due to the COVID-19 pandemic required in-person programming (including MiM®) to close, creating an urgent need to migrate in-person programs and services to virtual delivery. Initially the migration to a virtual MiM® was done ad hoc by local Alzheimer Societies. To ensure a safe and effective virtual MiM®, however, the program was standardized and evaluated. The purpose of this manuscript is to describe virtual MiM® and evaluate its feasibility and effectiveness.

## Materials and methods

This is a mixed method (Quant.+Qual., convergent design), pre-post study of the virtual MiM® program among people living with dementia and care partners. The evaluation was conducted among virtual MiM® participants at six Alzheimer Societies in Ontario from February 2022 to January 2023. Quantitative assessments were conducted in the two-weeks prior to the program start (pre-program) and within two weeks of program completion (post-program). Semi-structured interviews occurred with a subset of participants and all willing MiM® instructors. This study was approved by a University of Waterloo Ethics Committee (ORE: 43636). All participants provided verbal informed consent over Zoom® or telephone. Within the consent process, participants could indicate whether they were willing to complete an interview if invited.

## Participants

People living with dementia and their care partners were identified through the participating Alzheimer Societies (convenience sampling). They were eligible for the study if they were residents of Ontario (Canada), determined suitable for MiM® by Alzheimer Society staff, and willing to participate in both virtual MiM® and the evaluation assessments. People with mild to moderate dementia and their care partners who were screened safe to exercise were considered suitable for MiM®. Safety for exercise was screened using the Get Active Questionnaire [40]. Approval from a health care professional was elicited if indicated to ensure stability of medical conditions and safety to exercise. New MiM® participants were given priority for program and study enrolment. If a program was not full, participants could re-enroll in the program; however, participants only completed the research assessments once.

## Virtual Minds in Motion®

MiM® is an 8-week program for individuals with mild to moderate dementia and their care partners, and includes one 90-minute session weekly. Each session included a 5-minute welcome and safety review, 40 to 45min of exercise, and 30 to 45min of cognitive stimulation and social engagement. All sessions were completed live through a Zoom® web-conference platform. Classes included a maximum of 16 participants (8 people living with dementia and 8 care partners). Participants were required to have webcams on for safety, with microphones active during the cognitive stimulation and social engagement component.

Exercise was led by a qualified exercise leader (for example, registered Kinesiologist, clinical exercise physiologist, or person with Seniors' Fitness Instructor Certification). There was at least one additional staff or volunteer, and most often two. The additional staff/volunteer(s) demonstrated the seated version of exercises and provided tech support and watched for signs of physical distress and safety concerns.

As part of program intake (that is, not for the evaluation) participants report contact information, emergency contact information, medical history, and the exercise screening form (Get Active Questionnaire). Participants also completed a functional assessment that included the 5-timed sit-to-stand test and a tandem and semi-tandem stand test while being observed by the certified exercise instructor. Based on the exercise instructor's subjective assessment of participant performance, instructors advised participants to complete the program standing (if safe) or sitting.

Each exercise session included aerobic exercise, resistance training, and balance training with a goal of reaching moderate intensity. Warm-up and cool down was light aerobic exercise. Resistance exercises were done with body weight or small weighted household goods (for example, filled water bottles, canned goods, or small weights), as available. For most exercises, two levels of exercise were demonstrated, a seated version and a standing version.

Cognitive stimulation and social activities were facilitated by an Alzheimer Society staff member. Activities aligned with those used for cognitive stimulation therapy [36, 38]. Activities focused on stimulating thinking, memory, and social interaction and involved problem solving, mindfulness, and reminiscence. Instructors encouraged participants to share their lived experiences in relation to dementia and facilitated social relationships between participants.

## Assessments

The research assessment was completed by a research assistant pre-program and post-program to assess well-being, physical activity levels, and cognitive leisure activity levels. While people living with dementia independently completed the dementia quality of life questionnaire, their

care partners remained in the room when they completed the International Physical Activity Questionnaire and the Cognitive Leisure Activity Scale. This ensured care partners could provide their input if the individual living with dementia was having trouble recalling their activities over the past week. At post-program, satisfaction questionnaires were completed by all care partners and by people living with dementia who were able to do so. Semi-structured interviews were also completed at post-program by a subset of participants. All research assessments were completed on Zoom®. Assessments were completed separately and consecutively by the person living with dementia and their care partner, except when the person living with dementia preferred their care partner to remain nearby.

**Demographics.** Participants reported age (years), gender (*male*, *female*, *non-binary*, *prefer not to say*, *other*), city of residence, ethnicity, primary language spoken (*English*, *French*, *other*), education (*<high school, high school, technical or apprenticeship training, college or university degree, post-graduate studies*), household income (*<$25,000, $25,000-$49,000, $50,000-$99,999, $100,000-$149,999, greater than $150,000, prefer not to say*), marital status (*never married, married, partner/significant other, widowed, separated/divorced*), and type of residence (*private home, condo/apartment, seniors residence*). Care partners also reported their relationship to the person with dementia (*spouse/common law partner, family member, friend, other*). Participants reported town/city of residence was used to determine whether the participant lived in a rural area (population <1000), small population centre (population between 1,000 and 29,999), medium population centre (population between 30,000 and 99,999), or large population centre (population >100,000) [41].

**Feasibility.** *Completion rates*. The portion of participants completing the program and assessments were noted, along with reasons for drop out.

*Attendance and Intention to re-enrol*. Instructors recorded attendance of each participant. During the post-program assessment, the participants reported if they re-enrolled in the program or if they intended to re-enrol.

*Program satisfaction*. All care partners and people living with dementia (who were able) completed a program satisfaction questionnaire at post-program. Participants responded to 13 questions related to their experience in the program, effectiveness of instructors, difficulty of program, and perceived benefits of the program. For 11/13 questions, participants could select from 5 answer choices which ranged from 'strongly disagree' to 'strongly agree'. For the question, '*How would you rate your overall experience in the program*?', participants could respond with: *poor* (1), *fair* (2), *good* (3), *very good* (4), or *excellent* (5). For the question, '*How would you rate the difficulty of the program*?', participants could respond with: *too hard* (5), *somewhat hard* (4), *just right* (3), *somewhat easy* (2), and *too easy* (1) [S2 File].

*Fidelity*. A total of 9 fidelity observation checks were completed during the evaluation across six Alzheimer Society in Ontario that were involved in delivery. The fidelity observation checklist was created through collaboration between researchers, Alzheimer Society staff, and an advisory committee, and was completed by a research assistant. The checklists documented fidelity to intended program practices, which included instructor/volunteer qualifications, practices used during physical activity and cognitive stimulation, and interactions between instructors/volunteers and participants [S3 File].

**Effectiveness.** *Well-being*. Both people living with dementia and their care partners reported the perceived impact of the program on overall well-being, as well as sense of inclusion and belonging, using a 5-point Likert scale post-program ('*not beneficial at all*' to '*extremely beneficial*'). In addition, the Dementia Quality of Life Questionnaire (DEMQOL) and the DEMQOL-proxy were used to assess the health-related quality of life of people living with dementia [42]. Both questionnaires were designed and validated for use with individuals with a diagnosis of dementia. The DEMQOL questionnaire includes 28 questions, each

assessed on a four-point scale (*a lot*, *quite a bit*, *a little* and *not at all)*. The questions relate to five domains: health and well-being, cognitive functioning, daily activities, social relationships, and self-concept. Scores range from 28 (lowest quality of life) to 112 (highest quality of life). Participants also rate their overall quality of life, reported as *poor* (1), *fair* (2), *good* (3), or *very good* (4). The DEMQOL-proxy questionnaire includes similar questions but is answered by the care partner about their friend/family member living with dementia. Total scores can range from 31 (lowest quality of life) to 124 (highest quality of life), plus an overall rating of quality of life. DEMQOL has exhibited high reliability and moderate validity in individuals with mild to moderate dementia, whereas DEMQOL-proxy has demonstrated good acceptability and internal consistency, as well as high validity in people living with mild, moderate, and severe dementia [43, 44].

Three questionnaires were used to assess the well-being of care partners. The Warwick-Edinburgh Mental Well-Being Scale (WEMWBS) was used to assess their mental well-being [45]. The 14-item WEMWBS covers emotional, cognitive and evaluative dimensions, as well as psychological functioning. Participants select from 5 categories (*none of the time, rarely, some of the time, often,* and *all of the time)*. Higher total scores (14–70 points) indicate greater mental well-being [46]. The Social Provisions Scale was used to assess the social well-being of care partners. The Social Provisions Scale is an accurate predictor of social support, psychological distress, and overall quality of life among diverse populations, with good test-retest reliability (r = 0.86) and internal consistency reliability (alpha = 0.96) [47, 48]. Participants respond to 24 statements with *strongly disagree, disagree, agree,* or *strongly agree*. Total scores range from 24–96, where a higher score indicates greater perceived social support and well-being. The 12-item Short Form Health Status Survey (SF-12), a 12-item questionnaire derived from the SF-36 [49], was used to assess the physical and mental health-related quality of life of care partners. The two sub-scores (physical and mental component scores) each have a mean of 50 in the general U.S. population, with a one-point difference indicating 0.1 standard deviations [50].

*Physical activity*. Both people living with dementia and care partners reported the perceived impact of the program on their physical activity levels using a 5-point Likert scale post-program ('not beneficial at all' to 'extremely beneficial'). In addition, physical activity was reported pre- and post-program using the International Physical Activity Questionnaire Short Form (IPAQ-SF). The IPAQ-SF includes seven questions that measure vigorous activity, moderate activity, walking, and sitting over the last week and has demonstrated acceptable reliability (coefficients of 0.76–0.79) across age groups [51, 52]. IPAQ-SF was scored according to the standardized scoring guidelines [53], including truncation of the data above 3 hours per day of vigorous activity, moderate activity, or walking.

*Cognitive leisure activity*. Participants reported cognitive leisure activity using the Cognitive Leisure Activity Scale (CLAS), which is a validated assessment of types and frequency of cognitive activities [54]. Participants report how often they engage in 16 cognitive activities. For this study, the scale was modified to capture the impact of COVID-19. While the cognitive activity probes remained the same, participants could respond according to five categories: *daily* (4), *several times per week* (3), *several times per month* (2), *several times per year* (1), *never* (0), or *not right now due to COVID* (0). The maximum total score was 64, where a greater score indicates higher levels of cognitive leisure activities.

**Semi-structured interviews.** Purposive sampling was used to select approximately one person living with dementia (total = 7) and one care partner (total = 8) per month to complete an interview, selected from among participants who indicated willingness to do an interview. Interviewees were selected to ensure diverse perspectives, including men and women and people of different ethno-cultural groups. In addition, all instructors who delivered the program at

participating sites, and were willing, completed an interview (total = 11). Interviews followed a group-specific semi-structured interview guide [S1 File]. Questions probed participants' and instructors' experience in the program, perceived impact of the program, challenges experienced, and recommendations to improve the program. Interviews took between 14 and 33 minutes.

## Analysis

**Quantitative analysis.** Quantitative analysis was completed with JASP statistical software (version 0.16.3; JASP Team, 2022). Baseline participant characteristics were described as mean (standard deviation) or percentage (number) by participant type (people living with dementia, care partners). Change in continuous outcomes that were reported by both people living with dementia and care partners (physical and cognitive activity levels) were analyzed using a two-way mixed ANOVA with a within-subject factor for time (pre-program vs. post-program), a between-subject factor for participant type (people living with dementia vs. care partners), and an interaction for time*participant type. Significant main effects were followed with paired comparisons using Tukey's test. Change in outcomes that were reported by only one group (people living with dementia or care partners) were analyzed with a paired t-test. Significance was set at $p < 0.05$.

**Qualitative analysis.** Semi-structured interviews were audio recorded and transcribed verbatim. Transcripts were analyzed using general thematic analysis [55]. In brief, researchers (BN, CX) read and re-read transcripts to familiarize themselves with the data. All transcripts were independently coded line-by-line by both researchers using QSR International's NVivo qualitative data analysis software (March 2020). Inductive coding was used for thematic analysis. Once coding was complete, codes were then sorted according to participants' and instructors' experiences, views, and recommendations until coherent themes emerged. As themes emerged, relevant quotes were gathered, and draft themes were defined. Draft definitions and associated quotes were circulated to other members of the research team (VB, CSD, LM) for reflection and themes were iteratively adapted until consensus was achieved on the finalized themes. An identification code follows each quote to identify the participant from whom the quote came, where 'PLWD-Pxxx' refers to a participant who was living with dementia, 'CP-Pxxx' refers to a participant who was a care partner, and 'Ixxx' refers to an instructor or volunteer.

## Results

### Participants

Two-hundred and three people (112 people living with dementia, 91 care partners) participated in 27 virtual MiM® programs offered by the six participating Alzheimer Societies. Of those who participated, 201 (111 people living with dementia and 90 care partners) were eligible for and agreed to participate in the evaluation and completed baseline assessments (Table 1). Overall, there were a total of 335 enrolments (196 people living with dementia and 139 care partners), which includes the enrolments of participants who enrolled in the program multiple times. Participants who enrolled in the program multiple times were not eligible to repeat the research assessments. The average age of people living with dementia was 78.0 ± 7.5 years, and for care partners was 70.4 ± 10.2 years. For people living with dementia, most participants were white, married, and had a post-secondary degree. For care partners, most participants were female, white, married, and had a postsecondary degree. A detailed breakdown of participants' demographic information is shown in Table 1.

**Table 1. Participant characteristics by participant type (people living with dementia, care partners), expressed as mean (standard deviation) or percent.**

| | People Living with Dementia (n = 111) | Care Partners (n = 90) |
|---|---|---|
| Age (years) | 78.0 ± 7.5 | 70.4 ± 10.2 |
| Sex (% female) | 48.6 | 75.6 |
| Ethnicity (%) | | |
| White | 83.6 | 90 |
| Black | 4.5 | 3.3 |
| South Asian | 4.5 | 1.1 |
| Other | 7.4 | 5.6 |
| Rural Status (%) | | |
| Rural | 5.4 | 3.3 |
| Small population centre | 25.2 | 25.6 |
| Medium population centre | 15.3 | 18.9 |
| Large population centre | 54.1 | 52.2 |
| Type of Residence (% living in private home) | 72.1 | 76.7 |
| Level of Education (% more than high school) | 73.0 | 83.4 |
| Current Marital Status (% single or widowed) | 29.7 | 6.7 |
| Total Household Income | | |
| Under $50,000 | 28.8 | 21.1 |
| $50,000-$100,000 | 26.1 | 40.0 |
| $100,000+ | 22.6 | 18.9 |
| Prefer not to say | 22.5 | 20.0 |
| Relation of CP to PLWD (% spouse/common law) | NA | 74.4 |

## Feasibility outcomes

Of the 201 people who completed the baseline assessments, 162 people (87 people living with dementia, 75 care partners) completed the MiM® program (80.6%); 155 (83 people living with dementia, 72 care partners) completed both the pre- and post-program assessments and were included in the evaluation (77.1%). Ten people completed the baseline assessment but did not start MiM®. Common reasons for dropping out of the program included lost interest (n = 10, >25% of dropouts), too busy (n = 5), hospitalization of care partner (n = 2), and death of person with dementia (n = 2). Drop-outs were amplified because when one person needed to drop out, their partner usually did too. Reasons for dropping out after program completion but prior to the post-program assessment included vacation (n = 4), lost contact (n = 2), and hospitalization of care partner (n = 1). Seven of nine staff and four of six volunteers who were involved in program delivery also participated in the post-program interview.

**Attendance and intention to re-enrol.** People living with dementia who completed the program attended 86% of classes (range: 25% to 100%), matching care partners who also completed 86% of classes (range: 25% to 100%). Of those who completed the program, 83% of people living with dementia and 89% of care partners expressed an intention to re-enrol in the program.

**Program satisfaction.** The program was rated highly by both people living with dementia and care partners. The highest rated response for both groups was for the statement '*I feel I was treated with respect while participating in virtual Minds in Motion®* ' (4.7/5 for people living with dementia, 4.9/5 for care partners). The lowest rated response for both groups (3.9/5 for people living with dementia, 4.0/5 for care partners) was in response to the statement '*I was*

*satisfied with the number of sessions per week*'. For the question, 'How would you rate the difficulty of the program?', people living with dementia responded with an average score of 3.0, while care partners average score was 2.8. Therefore, on average care partners rated the program as being slightly easier than did people living with dementia. Detailed program satisfaction results are shown in Table 2.

**Fidelity observation checklist.** Fidelity to the MiM framework was very high across all categories, with many achieving 100% adherence across observations (instructor and volunteer training and qualifications, program practices, cognitive stimulation and social engagement activities instructor top behaviour, volunteer top behaviours). Other criteria were all above 90% adherence across programs (safety 94.8%, program delivery 96.9%, class component–quality 93.9%, class component–participant responsiveness 93.8%, physical activity instructor top behaviours 98.2%).

## Effectiveness outcomes

**Well-being.** *People living with dementia*. Health-related quality of life as reported by either people living with dementia (DEMQOL) or by care partners (DEMQOL-proxy) increased over the MiM® program (P<0.006). (Detailed scores in Table 3). Most people living with dementia (63.0% reported the program as extremely or very to their well-being. Approximately two-thirds (68.5%) of participants living with dementia reported that the program was extremely or very beneficial to their sense of inclusion and belonging.

*Care partners*. Changes in care partner well-being were inconsistent. SF-12 mental component scores improved (p = 0.017) whereas other scores did not meet significance thresholds. (Detailed scores in Table 3). However, approximately two-thirds of care partners (67.6%) reported that the program was extremely or very beneficial to their well-being. Just over 70% of care partners reported that the program was extremely or very beneficial to the sense of inclusion and belonging.

**Table 2. Program satisfaction among people living with dementia (n = 54) and care partners (n = 68).** Higher scores indicate greater satisfaction.

| Statement/Question | People Living with Dementia (SD) (/5) | Care Partners (SD) (/5) |
|---|---|---|
| I enjoyed participating in the Minds in Motion® virtual program. | 4.5 (0.5) | 4.8 (0.4) |
| I feel that I have benefitted from the Minds in Motion® virtual program. | 4.4 (0.6) | 4.6 (0.6) |
| I feel I was treated with respect while participating. | 4.7 (0.5) | 4.9 (0.4) |
| I feel that the program leaders did a good job leading the program. | 4.6 (0.5) | 4.9 (0.3) |
| I felt comfortable asking questions or sharing concerns with the staff. | 4.3 (0.6) | 4.7 (0.5) |
| I enjoyed the physical activity part of the Minds in Motion® virtual program. | 4.4 (0.6) | 4.8 (0.4) |
| I enjoyed the cognitive stimulation and social engagement part of the Minds in Motion® virtual program. | 4.3 (0.8) | 4.6 (0.7) |
| I felt comfortable taking the program in a virtual setting. | 4.5 (0.5) | 4.8 (0.4) |
| I was satisfied with the length of the class. | 4.3 (0.7) | 4.5 (0.7) |
| I was satisfied with the number of sessions per week. | 3.9 (1.0) | 4.0 (1.1) |
| I would recommend the Minds in Motion® virtual program to other people. | 4.5 (0.5) | 4.9 (0.4) |
| How would you rate your overall experience in the program? | 4.0 (0.9) | 4.5 (0.7) |

**Table 3. The pre- and post-program well-being outcomes for people living with dementia (PLWD, n = 78) and care partners (CP, n = 69).**

| Measure | Pre-Program Mean (SD) | Post-Program Mean (SD) | p-value for difference |
|---|---|---|---|
| **People Living with Dementia** | | | |
| Quality of Life (DEMQOL) (/112) | 88.0(12.1) | 91.0(11.3) | 0.003 |
| Quality of Life, poxy-reported (DEMQOL-Proxy) (/124) | 93.5(13.2) | 96.9(11.1) | 0.006 |
| **Care Partners** | | | |
| Mental Well-being (WEMWBS) (/70) | 51.5(7.5) | 52.4(6.7) | 0.059 |
| Social Well-being (Social Provisions Scale) (/96) | 79.9(9.9) | 80.6(10.2) | 0.154 |
| Physical health-related quality of life (SF-12 Physical Component Score) | 45.5(9.7) | 45.1(8.9) | 0.353 |
| Mental health-related quality of life (SF-12 Mental Component Score) | 44.0(10.0) | 46.6(9.4) | 0.017 |

*Note.* DEMQOL = Dementia Quality of Life Questionnaire; WEMWBS = Warwick-Edinburgh Mental Well-Being Scale; SF-12 = 12-Item Short Form Survey.

**Physical activity.** The two-way mixed model ANOVA of physical activity (IPAQ-SF) showed a main effect of time ($F_{(1,148)}$ = 11.01, p = 0.001) and a main effect for participant type ($F_{(1,148)}$ = 14.41, p<0.001) but no significant interaction between these factors ($F_{(1,148)}$ = 0.576, p = 0.449). In paired comparisons, physical activity levels improved from pre- to post-MiM (1649.0(2119.8) weekly METs versus 2141.8(2227.0) weekly METs, respectively, p = 0.001). In addition, physical activity levels were higher among care partners (2235.2 (2685.8) versus 2855.8(2713.1) weekly METs in care partners, compared to 1175.9(1360.9) versus 1565.4(1527.8) weekly METs in people living with dementia). In addition, 57.4% of people living with dementia and 69.1% of care partners described the MiM[®] program as extremely or very beneficial for improving their physical activity levels.

**Cognitive leisure activity.** The two-way mixed-model ANOVA of cognitive leisure activity indicated a main effect of time ($F_{(1,144)}$ = 52.04, p<0.001) and a main effect of participant type ($F_{(1,144)}$ = 42.179, p<0.001), but no interaction between time and participant type ($F_{(1,144)}$ = 0.398, p = 0.529). In paired comparisons, cognitive leisure activity levels increased from pre- to post-program (24.0(6.4) versus 26.6(6.6), respectively, p<0.001).

## Semi-structured interviews

Thematic analysis of semi-structured interviews among participants and instructors identified three over-arching themes: 1) social connection through virtual programs is possible, though challenges exist; 2) virtual MiM[®] improves abilities and motivation within and beyond the program, and; 3) virtual programs are convenient and feasible, depending on the person.

**Social connection through virtual program is possible, although challenges exist.** The theme that social connection could be created through the virtual MiM[®] program was consistent across interviews with people living with dementia, care partners, and instructors. They were able to identify challenges but also strengths of virtual programs in forming social connections. Interacting virtually was novel to many participants, but they were able to adjust to this format with time. One participant noted, "*It's a little–yes a little weird for me, and my daughter does a lot of Facetime too. But I've gotten really used to it now, so I liked it. And now a days there's not really any other way of doing it anymore*" (PLWD-P091).

Both instructors and participants felt that they were able to create a sense of community and belonging within the MiM[®] program. Participating in a program with others who were on a similar journey helped create connection and community among participants. Several participants noted that they felt understood and that they learned from one another. One participant noted, "*I think that, you know, talking to people and seeing what other people are doing*

*it's always good. Because then you can look at things and think, "I think I can do that and it would probably be of some benefit to me"* (PLWD-P116). Connection among participants was evident from their desire to interact with each other and to socialize even after the program ended for the day. One participant described how they enjoyed staying on Zoom® after the program was supposed to end, "*Like when we went in–you know, the last two times, the best time was after the program was supposed to have ended. So, when it was like 2:30, then some of us would just stay on longer*" (CP-P118).

The unique interactions that occur in a virtual environment helped to further establish a sense of community. In the virtual program, all participants were joining from their homes. Other participants could observe the random events that occurred in other participants' backgrounds. This offered an opportunity to learn about their family and life. One volunteer noted, "*Mostly [participants and instructors] were doing it from home and so there's you know pictures on the wall behind or furniture or a dog running through. But you sort of–to me that just gave it all a bit of flavour so I didn't mind it*" (I009). Also, in the virtual environment, conversations were shared among all participants and instructors rather than in smaller groups, creating connection among the group as a whole. Shared events, even when challenging, could create a sense of shared experience. One instructor noted, "*Well there were a couple of times when we had tech issues that had nothing to do with either the participants or the facilitators. You know I think there was one time when we all got kicked out, like the recovery for the link got changed like by somebody else in the organization. . . And so this mysterious like what! Obviously that's challenging. But, you know, again the beauty and satisfaction of it is that we all experienced it together as a group*" (I005).

Despite the sense of community created within the virtual MiM program, there was acknowledgement that the virtual social environment was different than in-person. Several participants and instructors noted that it was impossible to have a one-on-one conversation with another person without everyone else listening, noted as a strength (above) but also as a weakness. An instructor commented that, "*if they've decided that they both live in Sudbury they'll just connect on that level. But it's not the same because they feel like they're talking, you know, having this conversation while everyone else is still listening. It's different than in person for sure*" (I003).

**Virtual MiM® improves abilities and motivation within and beyond the program.** Participants described the impact of the program on their physical abilities and motivation to be physically active. They recognized improvements in their physical abilities as they could do tasks they could not do prior to the program. One participant described being able to get through the entire exercise component without any breaks, which he couldn't do the first few sessions. He noted, "*I know I felt like my strength in general has really declined since I stopped working at the beginning of COVID. And I've noticed, even though I think the intensity of the classes has increased over time. . . I'm feeling stronger now and able to get through just fine compared to the beginning*" (CP-P092). Another participant reflected on their physical function as they entered the program, but also on the improvements they experienced, "*Like the first couple of weeks with the chair, getting up off your chair, I think we did it about 15 times, whatever [the instructor] was doing, I realized that I had lost strength in my legs. But the more we did it, then the better we got*" (CP-P007).

Participants felt motivated to be active as a result of virtual MiM®. Participants noted that they now recognized the benefits of physical activity and wanted to stay active in the future. One participant said, "*. . . I realized the benefit of taking time to do a program of some description for half an hour. . . It's made me more aware that you have to make the time to do this because it's what your body needs. You have to look at the body like an engine, you have to maintain it. It's made me more aware like that*" (CP-P007). Another described how physical activity

had been recommended to them but that their participation in MiM really augmented their motivation, "*I think that's something we thought of based upon the involvement in what we were doing. And it's kind of a little bit of a bee in my butt. . . it got me thinking we need to do more of that*" (CP-P118).

**Virtual programs are convenient and feasible, depending on the person.** This third theme suggested that virtual programs were not only feasible but also convenient in many ways, though comfort with and access to technology may be a barrier for some. Participants described how the virtual MiM® program can help overcome barriers to participation, including transportation (time and cost), weather conditions, lack of time, and risk of contracting COVID-19. Describing the advantage of the virtual program over the in-person MiM®, one participant said, "*If they stop doing it online and start saying, "Ok, show up at this arena on the other side of town at 10:00 instead" . . . it's just not as attractive as being able to just do it at home. And then shower immediately and get changed and then get onto something else, it takes less time, less gas*" (CP-P092).

Both instructors and participants also described how the program created a sense of routine to participants' week during the COVID-19 pandemic. At the time of the virtual MiM sessions, many participants were still careful about leaving their homes as they wanted to limit their exposure to COVID-19. In this setting, they noted that the program was often something that they looked forward to and had become a staple in their weekly agenda. A care partner, speaking about her husband living with dementia noted, "*he looked forward to it and would remember that it was on a Wednesday morning*" (CP-P145). Another participant spoke about the purpose it brought to their week, "*[The program is] important because you have a purpose. Your day has a purpose*" (CP-P007).

Despite strengths of virtual MiM®, there were challenges to virtual programming. Speaking of the demographic of older adults and their challenges with technology, one instructor noted, "*it's a whole learning thing to press the right buttons. . ., [to] enlarge the screen, turn the mute off. . . They may have a different computer, they may have less experience and so that is challenging*" (I005). There was a perception among participants and instructors that tech supports need to be a fundamental part of virtual MiM® (and other virtual) programs. Because of this, instructors suggested having a dedicated volunteer/instructor to provide technology support, as it was not possible for instructors to simultaneously lead the class and provide tech support. Despite the challenges of supporting technology from afar, no instructors thought it was detrimental to the delivery of the program.

There was also a sense that participants' quality of experience may depend on their available technology and how well it was suited to their needs. Many participants found it easy to join the program from their tablets or laptops, which usually had a built-in microphone and webcam. One participant noted specific challenges seeing a small laptop screen, "*It was so small. And we have a–this is supposed to be a 15-inch laptop, and people were using tablets. So I can't imagine how small we appeared*" (CP-P118). Others described difficulties hearing the music during exercise or the voices of those speaking. Some participants found solutions to these challenges by connecting their computer to their TV in order to enlarge their screen and make their volume louder/clearer, although one's ability to do this would have been dependent on their own technology skills.

## Discussion

This study used a mixed-methods study design to evaluate virtual MiM® among people living with dementia and care partners, with additional insights from instructors. People living with dementia reported significant increases in health-related quality of life, whereas care partners reported improvements in mental well-being but not in other domains of well-being or quality

of life. Participants as a whole, reported increased physical activity and cognitive leisure activity levels after MiM®. Quantitative results were complemented by qualitative results that indicated that virtual MiM® was a convenient, feasible option that improved physical abilities and created social connections among participants.

Improvements in physical activity and cognitive leisure activity after MiM® were consistent between people living with dementia and care partners. This finding is in line with the evaluation of in-person MiM® [39], which also found improvements in physical activity levels (but did not assess cognitive leisure activity). While other online exercise programs for people living with dementia and their care partners have also shown the ability to moderately increase participants' physical activity levels post-program [56], there are no other studies with online programs that reported on participants' cognitive leisure activity levels pre- and post-program. The results here may be somewhat confounded by changes in COVID-19 restrictions over the duration of our study, where fewer restrictions were in place by the end of our evaluation period. In post hoc analyses, fewer people reported that they were not participating in cognitive leisure activities due to COVID-19 at the post-intervention. However, further analyses that eliminated any '*not now due to COVID-19*' responses still showed a significant increase in cognitive leisure activities at post-program ($p<0.01$).

Improvements in quality of life were notable among people living with dementia, whether self-reported or reported by their care partners. These results largely align with the evaluation of in-person MiM®, which used only the WEMWBS to assess mental well-being and found improvements among people living with dementia [39]. One other study has reported a trend towards an improvement in quality of life in people living with dementia, after they completed a 12-week, group exercise program through Zoom®, although the trial was too small to report statistical significance [56]. However, among older adults in general, several online exercise programs have reported improvements in quality of life and mental health of participants [57, 58].

Changes in the well-being of care partner participants somewhat inconsistent across domains but showed positive trends for improved mental well-being, with improvements in mental health-related quality of life (SF-12 Mental Component Score) (p = 0.002) and nearing statistically significant improvements in mental well-being on the WEMWBS (p = 0.059). Again, this finding is similar to the in-person MiM®, which also found improvements on the WEMWBS (the only assessment of well-being) [39]. While mental well-being appeared to improve, changes physical and social dimensions of well-being among care partners did not reach statistical significance. This is in contrast to improvements in objective assessments of physical function (endurance and strength) with the in-person MiM® [39]. There are several possible explanations. First, it is possible that the physical demands of virtual MiM® were not sufficient to improve their physical well-being, though care partners generally described the difficulty of the program as about right. Alternatively, it is possible that the oversight they provided to their partner living with dementia in virtual MiM® meant that it was more difficult for them to participate and improve physically or develop social connections. Furthermore, the quality of life of care partners decreases with the severity of dementia among the person they are caring for and has been reported to be worse during the COVID-19 pandemic [59–61]. It is possible that the program was insufficient to overcome this impact. Finally, it is also possible that the assessments used were not appropriate to this group. While we had proposed to use the WHO-QOL, a validated measure of quality of life across diverse age, gender, and education levels [62], our advisory group found some of the questions objectionable. As a result, we used a variety of assessments that captured different elements of quality of life and well-being but they may have been less sensitive as a whole. It should be noted that care partners reported that they perceived benefits to their overall well-being despite a lack of statistical improvement in questionnaires.

Despite the virtual environment, people living with dementia and care partners felt they developed a sense of community and social connections with other participants. This aligned closely with the results of a recent qualitative study that analyzed the impact of virtual memory cafés delivered monthly for people living with dementia and their care partners, in which participants reported a feeling of belonging to the group and a sense of community that formed between attendees [63]. This is further supported by other virtual, education, social, and fitness programs among older adults that have shown to be positively associated with feelings of connection and better mood, and lower feelings of social isolation [64]. Another study extended these findings to indicate that feelings of social connection developed online were positively associated with quality of life [65]. Therefore, it's possible that improvements to the well-being of participants was also facilitated by feelings of social belonging and connection received through the program.

Overall, virtual MiM® was an acceptable, feasible, and safe option for exercise and cognitive stimulation among people living with dementia and their care partners. Satisfaction levels were very high across all criteria, and over 80% planned to re-enrol in the virtual MiM®, despite programs being available in-person for most of the evaluation period. Evidence has suggested that adherence to live, technology-based programs tends to be greater than in-person programs [66], with adherence highest when completing the program with a family member or friend, while joining from their home [25, 67]. This is supported by this study's high attendance rate and low drop-out rate compared to the previous evaluation of the in-person MiM® program, where the virtual MiM® program had higher attendance rates (86% versus 80%) and lower program drop-out rates (19% versus 25%) [39]. The virtual MiM® program also appears to be safe. No adverse events occurred across the six participating sites. Similarly, other programs have found that virtual programming was a safe option during the COVID-19 pandemic [68]. By including care partners, the ability to comply with safety protocols if an adverse event was to occur was enhanced, as implemented in and recommended by several other studies [68, 69]. Our results suggest that virtual programming should continue to be made available regardless of the prevalence of COVID-19, especially given the multitude of barriers that exist for in-person programming.

Our study does have some limitations. First, this was a pre/post evaluation of the virtual MiM® program. It's not possible to know conclusively whether observed changes (or stability) in outcomes are due to the program, disease progression, repeated assessment, or other factors. However, controlled trials have demonstrated that exercise and cognitive stimulation provide benefits to people living with dementia [70, 71]. MiM® includes physical and cognitive stimulation and it was not possible for us to evaluate the independent contributions of these components. In addition, assessing physical activity levels in a country with four seasons can be difficult, as physical activity levels fluctuate considerably due to seasonal variation in weather [72]. However, the evaluation ran from February 2022 to January 2023 and so changes due to seasons likely balanced out. In addition, participants tended to be white, have high levels of education and were more likely to be married than the general population. Additional actions should be considered to improve access to community programs like MiM® for individuals who are of a non-white ethnicity, are single/widowed, and have lower levels of education and income. Finally, satisfaction questionnaires were only completed by participants that completed the entire program and may be positively biased.

## Conclusion

This study found the virtual MiM® program for people living with dementia and care partners to be acceptable, safe, and effective at improving the overall quality of life of people living with dementia and the mental well-being of care partners. Furthermore, the virtual MiM® program

is effective at improving the physical and cognitive activity levels of participants. Broadly, virtual programs have the potential to create social connections among participants, while being convenient to attend and impactful. The virtual MiM® program continues to run throughout Ontario, Canada.

## Supporting information

**S1 File. Semi-structured interview guide.**
(DOCX)

**S2 File. Program satisfaction questionnaire.**
(DOCX)

**S3 File. Fidelity observation checklist.**
(DOCX)

**S1 Data.**
(ZIP)

## Acknowledgments

The authors would like to acknowledge and thank Minds in Motion® instructors from the Alzheimer Society of Ontario who delivered the program and supported data collection. The authors would also like to thank all participants who participated in the research study, and Simran Bansal who assisted with several research assessments.

## Author Contributions

**Data curation:** Bobby Neudorf.

**Formal analysis:** Bobby Neudorf.

**Funding acquisition:** Christina Stergiou-Dayment.

**Investigation:** Bobby Neudorf, Laura Middleton.

**Methodology:** Bobby Neudorf, Christopher Dinh.

**Project administration:** Bobby Neudorf, Vanessa Barnes.

**Supervision:** Laura Middleton.

**Writing – original draft:** Bobby Neudorf.

**Writing – review & editing:** Bobby Neudorf, Vanessa Barnes, Christina Stergiou-Dayment, Laura Middleton.

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
