## [Decision Letter · Decision Letter 0]

18 Jul 2023

PONE-D-23-07761Enhancing Minds in Motion as a virtual program delivery model for people living with dementia and their Care PartnersPLOS ONE

Dear Dr. Neudorf,

Thank you for submitting your manuscript to PLOS ONE. After careful consideration, we feel that it has merit but does not fully meet PLOS ONE’s publication criteria as it currently stands. Therefore, we invite you to submit a revised version of the manuscript that addresses the points raised during the review process.

 Interesting piece of work.  However, there are few methodological challenges you need to overcome.I understand this is a pre-post study to assess the effectiveness of intervention. 1. Why did you decide to do pre-post assessment before and after 2 weeks ? Pre-assessment I agree, however for the post assessment, Is there any justification of selecting two weeks interval?Also, I don't think you could conclude based on one post assessment soon after intervention. Any plans to assess the sustainability?In programme or project evaluation, there should be criteria for evaluation and also agreed timeline. You mention that you find out acceptability and impact on QOL. Did you think about the effectiveness of the intervention? Also, I think it's too early to do the post assessment in two weeks.DEMQOL which you used for QOL measurements has several components, and also its a complex measure of QOL which gives overall well being. QOL is a psychological construct which is multifactorial. Do you think about the other confounding effects when selecting the sample ? 

We look forward to receiving your revised manuscript.

Kind regards,

Surangi Jayakody, MBBS, MSc, MD

Academic Editor

PLOS ONE

Journal Requirements:

2. Please update your article type to ‘Research Article’.

Reviewers' comments:

Reviewer's Responses to Questions

**Comments to the Author**

1. Is the manuscript technically sound, and do the data support the conclusions?

Reviewer #1: Yes

Reviewer #2: Partly

Reviewer #3: No

2. Has the statistical analysis been performed appropriately and rigorously? 

Reviewer #1: I Don't Know

Reviewer #2: Yes

Reviewer #3: No

3. Have the authors made all data underlying the findings in their manuscript fully available?

Reviewer #1: Yes

Reviewer #2: Yes

Reviewer #3: Yes

4. Is the manuscript presented in an intelligible fashion and written in standard English?

Reviewer #1: Yes

Reviewer #2: Yes

Reviewer #3: Yes

5. Review Comments to the Author

Reviewer #1: I am interested in your topic because of it is very interesting. I have found some issues regarding your manuscript that needs to be addressed.

1. In the methods part, you have used mixed method(Quant.+Qual.). You better to specify which type of mixed method you have used(Explanatory sequential, exploratory sequential or concurrent mixed method etc)

2. You have used purposive sampling method to select your participants, you have to specify which type of purposive sampling method you used.

Minor comments:- You better to cite the tables and figures in your text in accordance with plos guideline(Tab1, Tab2 etc, Fig1, Fig2 etc.

Reviewer #2: This is an interesting interventional study. However some of the information needs clarification and those are included as comments in the attached document . Titles of Table 1 and 2 need to be revised.

Reviewer #3: Reviewer comments: Enhancing Minds in Motion as a virtual program delivery model for people living with dementia and their Care Partners

Overview:

It is an interesting study no doubt on an important aspect of supporting people living with dementia. However, its methodology is not sound enough to be worthy of publication in PLOS.

Methodology:

The study is entirely based on subjective reporting both quantitatively and qualitatively. Furthermore, we are not sure how patients with dementia were able to rate them accurately. We are not sure whether the study was powered adequately to reach a statistical significance.

There was no attempt at randomization or a control group to compare with. The sample was a convenient one and there were many non responders. They may not have responded as they did not benefit from the program.

Findings: Only the qualitative findings may have some value. I am not sure whether this is adequate for a publication in PLOS.

Thank you

6. PLOS authors have the option to publish the peer review history of their article (what does this mean?). If published, this will include your full peer review and any attached files.

Reviewer #1: **Yes: **Agmas Wassie Abate

Reviewer #2: No

Reviewer #3: No

---

## [Author Response · Author response to Decision Letter 0]

1 Aug 2023

Please see the attached document 'Response to Reviewers' to see our responses to each of the comments given by reviewers.

---

## [Editor Report · Decision Letter 1]

24 Aug 2023

Enhancing Minds in Motion as a Virtual Program Delivery Model for People Living with Dementia and Their Care Partners

PONE-D-23-07761R1

Dear Dr. BOBBY,

We’re pleased to inform you that your manuscript has been judged scientifically suitable for publication and will be formally accepted for publication once it meets all outstanding technical requirements.

Kind regards,

Surangi Jayakody, MBBS, MSc, MD

Academic Editor

PLOS ONE

Additional Editor Comments (optional):

PLEASE MAKE SURE THAT  YOU REMOVE THE IN-TEXT CITATION FROM THE ABSTRACT.

---

## [Editor Report · Acceptance letter]

29 Aug 2023

PONE-D-23-07761R1 

Enhancing Minds in Motion as a virtual program delivery model for people living with dementia and their Care Partners 

Dear Dr. Neudorf:

I'm pleased to inform you that your manuscript has been deemed suitable for publication in PLOS ONE. Congratulations! Your manuscript is now with our production department. 

Kind regards, 

on behalf of

Dr Surangi Jayakody 

Academic Editor

PLOS ONE